# Trace Amine Associate Receptor 1 (TAAR1) as a New Target for the Treatment of Cognitive Dysfunction in Alzheimer’s Disease

**DOI:** 10.3390/ijms23147811

**Published:** 2022-07-15

**Authors:** Damiana Leo, Giorgia Targa, Stefano Espinoza, Agnès Villers, Raul R. Gainetdinov, Laurence Ris

**Affiliations:** 1Department of Neuroscience, Research Institute for Health Science and Technology, University of Mons, 20 Place du Parc, 7000 Mons, Belgium; damiana.leo@umons.ac.be (D.L.); agnes.villers@umons.ac.be (A.V.); 2Department of Pharmacological and Biomolecular Sciences, University of Milan, Via Balzaretti 9, 20133 Milan, Italy; giorgia.targa@unimi.it; 3Central RNA Laboratory, Istituto Italiano di Tecnologia (IIT), 16163 Genova, Italy; stefano.espinoza@iit.it; 4Institute of Translational Biomedicine, St. Petersburg State University, Universitetskaya Emb. 7-9, 199034 St. Petersburg, Russia; gainetdinov.raul@gmail.com; 5St. Petersburg University Hospital, St. Petersburg State University, Universitetskaya Emb. 7-9, 199034 St. Petersburg, Russia

**Keywords:** trace amine-associated receptor 1 (TAAR1), trace amine, β-amyloid, Alzheimer’s disease, glutamate receptors, cognitive impairments

## Abstract

Worldwide, approximately 27 million people are affected by Alzheimer’s disease (AD). AD pathophysiology is believed to be caused by the deposition of the β-amyloid peptide (Aβ). Aβ can reduce long-term potentiation (LTP), a form of synaptic plasticity that is closely associated with learning and memory and involves postsynaptic glutamate receptor phosphorylation and trafficking. Moreover, Aβ seems to be able to reduce glutamatergic transmission by increasing the endocytosis of NMDA receptors. Trace amines (TAs) are biogenic amines that are structurally similar to monoamine neurotransmitters. TAs bind to G protein-coupled receptors, called TAARs (trace amine-associated receptors); the best-studied member of this family, TAAR1, is distributed in the cortical and limbic structures of the CNS. It has been shown that the activation of TAAR1 can rescue glutamatergic hypofunction and that TAAR1 can modulate glutamate NMDA receptor-related functions in the frontal cortex. Several lines of evidence also suggest the pro-cognitive action of TAAR1 agonists in various behavioural experimental protocols. Thus, we studied, in vitro, the role of the TAAR1 agonist RO5256390 on basal cortical glutamatergic transmission and tested its effect on Aβ-induced dysfunction. Furthermore, we investigated, in vivo, the role of TAAR1 in cognitive dysfunction induced by Aβ infusion in Aβ-treated mice. In vitro data showed that Aβ 1–42 significantly decreased NMDA cell surface expression while the TAAR1 agonist RO5256390 promoted their membrane insertion in cortical cells. In vivo, RO5256390 showed a mild pro-cognitive effect, as demonstrated by the better performance in the Y maze test in mice treated with Aβ. Further studies are needed to better understand the interplay between TAAR1/Aβ and glutamatergic signalling, in order to evaluate the eventual beneficial effect in different experimental paradigms and animal models. Taken together, our data indicate that TAAR1 agonism may provide a novel therapeutic approach in the treatments of disorders involving Aβ-induced cognitive impairments, such as AD.

## 1. Introduction

Alzheimer’s disease (AD) is the most common cause of dementia, accounting for about 70% of causes [1]. AD progresses insidiously and continuously with three broad phases: preclinical AD, mild cognitive impairment and dementia. One of the main consequences of AD is the reduction of synaptic plasticity in brain regions involved in high-level cognitive functions, such as the hippocampus and the prefrontal cortex. The major neuropathological hallmarks of AD are the presence of extracellular amyloid plaques, intracellular neurofibrillary tangles of the microtubule-associated protein Tau and neuroinflammation [2]. Beta-amyloid peptide (Aβ) is elevated in the brains of AD patients and is believed to be causative in the disease process. The severity of memory impairments is highly correlated with Aβ amount since soluble extracellular Aβ appears to be responsible for the memory impairments before the accumulation of plaques [3]. The pathogenic Aβ forms are initiated by sequential cleavage of the amyloid precursor protein (APP) by enzymatic complexes known as beta- and gamma-secretases. Animal models with human APP overexpression or APP familial mutations showed an increase in Aβ production and severe impairments in synaptic transmission [4] associated with memory impairments. These phenomena precede neuronal death and plaque formation, indicating that alterations of key synaptic functions could contribute to the memory deficits associated with AD. Aβ influences both long-term potentiation (LTP), a form of synaptic plasticity that is closely associated with learning and memory, and long-term depression (LTD), an opposing form of synaptic plasticity [5]. LTP and LTD involve postsynaptic phosphorylation and glutamate receptor trafficking. In particular, it has been shown that amyloid can cause a reduction of glutamatergic transmission and the inhibition of synaptic plasticity via increased endocytosis of NMDA receptors resulting in reduced amounts of surface NMDA receptors in cortical neurons [6]. Finally, the direct intracerebral injection of Aβ peptides causes learning and memory deficits, as well as AD-like behavioural alterations [7].

Recently, a new “player” as a modulator of glutamatergic system came in: the trace amines-associated receptor 1 (TAAR1). Trace amines (TAs) are a family of endogenous compounds with strong structural similarities to classical monoamine neurotransmitters [8]. TAs such as β-phenylethylamine, tyramine, octopamine and tryptamine are produced from the same amino acid precursors as dopamine (DA), noradrenaline and serotonin by the aromatic-L-amino acid decarboxylase (AADC). TAs are present in mammalian tissues at nanomolar concentrations, at least two orders of magnitude below the levels of classical monoamine neurotransmitters. The molecular mechanism of the TAs involves binding to G protein-coupled receptors, called TAARs (trace amine-associated receptors) [9]. TAAR1, the most studied TAAR, is distributed in the CNS, mainly within monoaminergic systems. TAAR1 is expressed broadly in the brain, including the prefrontal cortex (PFC), hippocampus, ventral tegmental area and dorsal raphe [8,10]. TAAR1 can modulate the dopaminergic and serotonergic systems [11,12,13,14,15] and potentially affect glutamatergic activity [12,16]. It has been shown that the selective activation of TAAR1 by both full and partial agonists can reverse glutamatergic hypofunction induced by selective NMDA receptor antagonists in normal animals and mutant mice with NMDA receptor deficiency, suggesting that TAAR1 activation may enhance glutamatergic function [12,17]. Moreover, TAAR1 knockout (TAAR1-KO) mice displayed an altered subunit composition of cortical NMDA receptors and the dysregulation of NMDA receptor-dependent synaptic function, accompanied by increased perseveration and impulsivity. The activation of TAAR1 by specific agonists reduced impulsive behaviours in normal mice, further indicating a facilitating role of TAAR1 on cortical glutamate transmission and the associated behavioural functions [12]. Several lines of evidence suggested the pro-cognitive action of TAAR1 agonists in various behavioural experimental protocols. TAAR1 agonists could suppress the hyperlocomotion triggered by non-competitive NMDA receptor blockers phencyclidine and L-687414 [16,17]. Additionally, impulsivity in mice is decreased by TAAR1 activation [12,18] and TAAR1 promotes pro-cognitive effects in primates and rodents [19]. Recently, Wu and colleagues demonstrated that TAAR1 partial agonist RO5263397 significantly enhanced short-term memory retrieval [19]. There is a hypothesis that the reduction of dopamine and serotonin levels, disrupted calcium homeostasis and increased monoamine oxidase activity can contribute to AD pathology via defective trace amine metabolism [20]. Although there is no direct evidence of TAAR1 contribution to AD pathology at present, the modulation of the NMDA receptor [12,17] and activation of GSK3β due to TAAR1 signalling [21] suggest such possibility. Hence, targeting the TAAR1 receptor could provide a novel multifactorial approach for the treatment of AD [22].

In the current study, we investigated the ability of the TAAR1 agonist RO5256390, in vitro and in vivo, to counteract the Aβ effect. We demonstrated that TAAR1 agonists rescue the well-known Aβ-induced surface decrease of the NMDA receptor in mouse cortical cell cultures. The receptor surface expression analysis revealed that this effect is strictly TAAR1-dependent since TAAR1 KO cell cultures were insensitive to RO5256390 treatment. Furthermore, we demonstrated a behavioural correlate to this molecular effect: TAAR1 intracerebroventricular (i.c.v.) infusion ameliorates working memory in Aβ-treated mice, possibly acting on the glutamate receptor expression. These data could open up a new perspective in treating cognitive deficits in early AD phases. Hence, additional investigations are imperative before considering the TAAR1 modulator as a target therapy AD strategy.

## 2. Results

### 2.1. TAAR1 Agonists Rescue the Aβ-Induced Decrease of NMDA Subunits

To provide a mechanistic explanation for the modulatory role of TAAR1 over glutamatergic transmission, we analysed the surface expression of subunits of NMDA glutamate receptors following TAAR1 agonist application in cultured cortical neurons from WT and TAAR1-KO mice. We observed that Aβ regulates the surface expression of NMDA-type glutamate receptors (Figure 1 and Figure 2).

These data are consistent with recent findings demonstrating that Aβ reduces LTP and glutamatergic transmission [23,24]. Furthermore, 1 h of Aβ 1-42 treatment can reduce GluN1 receptor surface expression in cortical cells (14 DIV; six cultures, ANOVA *p* < 0.05; WT: Newman-Keuls multiple comparison test. WT “control vs. Ab 1-42” mean −100.6; q 6.620; “control vs. TAAR1 agonist” mean 99.50 q 10.8; “Ab 1-42 vs. Ab+Taar agonist” mean 36.12 q 6.353; KO: “control vs. Ab 1-42” mean 41.90; q 7.370. Figure 1) and promote the endocytosis of glutamate receptor subunits, as previously shown [25]. Aβ 1-42 was able to exert its effect in both TAAR KO and control animals (Figure 1), even if the receptor reduction was more evident in control animals. The subsequent addition of TAAR1 full agonist RO5256390 counteracted the GluN1 reduced expression selectively in cells from WT mice by restoring their expression to untreated levels (Figure 1). Similarly, RO5256390 had a peculiar activity on GluN1 receptor surface expression in cells from WT but not TAAR1-KO mice. We observed a comparable RO5256390-induced impact on GluN2A surface expression. As already described for the GluN1 receptor, we found a decrease in GluN2A surface levels after Aβ 1-42 medium application (Figure 2; 14 DIV; six cultures; ANOVA *p* < 0.05; WT: Newman-Keuls multiple comparison test. WT “control vs. Ab 1-42” mean 31.48; t 5.430; “control vs. TAAR1 agonist” mean −51.30 q 12.52; mean 41.90; “ Ab 1-42 vs. Ab + Taar agonist” mean −19.41 q 4.735; KO: “control vs. Ab 1-42” mean −31.68; q 9.685). RO5256390 could restore the receptor expression in cells from WT but not TAAR1-KO mice, demonstrating a specific TAAR1-mediated effect.

In an attempt to determine the molecular mechanisms regulating NMDA surface expression, we analysed different molecular players in NMDA receptor trafficking, such as postsynaptic density protein 95 (PSD-95), a modular protein that is enriched in postsynaptic density. In addition to their role in synaptic anchoring, the PSD-95 protein is important for the intracellular trafficking and synaptic delivery of NMDA receptors. Thus, we evaluated the PSD-95 protein levels in cortical cell cultures after TAAR1 agonist treatment. As is already known, Aβ caused the loss of the synaptic proteins PSD-95 and synaptophysin via NMDA receptor activation. Conversely, TAAR1 agonists demonstrated the ability to increase the PSD-95 amount (Figure 3; 14 DIV; three cultures, ANOVA *p* < 0.05; Newman-Keuls multiple comparison test “ctrl vs. Ab” mean −2.812; q 12.81).

### 2.2. TAAR1 Agonists Rescue Aβ-Induced Cognitive Deficits

Next, we focused on the in vivo model of AD. AD features were mimicked by the i.c.v. infusion of β-amyloid (1-42) peptide into the brain of WT and TAAR1-KO mice. Before Aβ fragment injection, we examined the assembly state of the soluble Aβ 1-42 preparation by WB analysis. We used β-amyloid after 24 h of aggregation at 37 °C. Immunoblot showed small-sized Aβ monomer, dimer and tetramer bands (Figure 4A), indicating that the predominant composition of the present soluble Aβ preparation were monomers and low size oligomers rather than fibrils.

We injected β-Amyloid i.c.v. and tested the effective distribution of the injected solution by using methylene blue. As shown in Figure 4B, the methylene blue diffused rapidly and we detected Aβ protein in the nearest injected brain areas (i.e., hippocampus, midbrain and cortex, latest not shown) (Figure 4B). We then employed the protocol shown in Figure 5 to perform in vivo studies using our Aβ mouse model.

We assessed the performance of treated mice in the Y maze task, which is widely used for evaluating memory in AD models and the open field test, providing various behavioural information ranging from the general ambulatory ability to data regarding the emotionality of the subject animal. Mice were divided in four groups: WT/KO first injection (Aβ or vehicle); for the second injection WT/KO Aβ received RO5256390 while WT/KO vehicle received a second vehicle injection (in order to minimise the possible effect of anaesthesia or manipulation). As shown in Figure 5 (n = 12 mice/group ANOVA Kruskal–Wallis; *p* < 0.05), we confirmed that Aβ infusion could induce a slight cognitive impairment in control animals, while the TAAR1-KO mice seem to be more resistant to Aβ effects. The second Y maze experiment was performed 1 week after the agonist injection. Figure 5 shows that, in this case, TAAR1 agonist rescued the memory deficit in control animals and they were able to perform Y maze test as the control ones.

As the total ambulatory distance between WT and TAAR KO mice was quite similar, we analysed the thigmotaxis, or the tendency of a subject to remain close to walls (Figure 6).

The degree of thigmotaxis is a measure of anxiogenic behaviour in mice [26], since it increases as anxiety levels rise. Using the Noldus EthoVision software, individual zones were overlaid on the paths travelled by the mice and time spent in inner zones versus outer zones was calculated and presented as a function of total time (10 min) in the maze. In this case, mice did not show anxiogenic behaviour. A representative travel path can be seen in Figure 6. The behaviour of the control and Aβ-treated animals in the OF test was similar in WT and TAAR1-KO mice. Thus, Aβ injection does not affect locomotor activity or induce anxiety-related behaviour.

## 3. Discussion

The present study demonstrated that the full TAAR1 agonist RO5256390 can increase cell surface NMDA glutamate receptor expression and counteract their Aβ-induced reduction in vitro. Furthermore, this effect is entirely dependent on TAAR1 agonism, as RO5256390 did not affect TAAR1-KO mice cortical cell cultures. The possible RO5256390 protective effect against Aβ was also revealed in vivo, as demonstrated by a better Y maze performance exclusively in WT Aβ-treated animals. These results are consistent with previous studies indicating that partial and full TAAR1 agonists promote a cognitive performance in rodents and monkeys [16,17,19]. We tried to mimic the early effects of Aβ on synaptic functions by using acute treatments and short time applications of Aβ in vitro; these also reflect slight differences between the different drugs treatments. More studies are needed to better understand the relationship between these molecular players.

Emerging recent evidence further supports the role of TAAR in neuropsychiatric disorders [8,9]. In mice that experienced chronic social defeat stress, a significant reduction in the level of TAAR1 mRNA was found in the medial prefrontal cortex [27]. In addition, chronic treatment with the selective TAAR1 partial agonist RO5263397 ameliorated chronic stress-induced changes in cognitive function, dendritic arborisation and the synapse number of pyramidal neurons in the medial prefrontal cortex. Finally, RO5263397 significantly enhanced the retrieval of short-term memory in control animals and the retrieval of long-term novel object recognition memory, confirming that those TAAR1 agonists have pro-cognitive properties.

Moreover, a new orally active compound, SEP-363856 (Ulotaront, TAAR1 agonist with 5-HT1A receptor activity), has been tested in recent clinical trials as a psychotropic agent for the treatment of psychosis in schizophrenia [28]. Four weeks of SEP-363856 treatment resulted in a greater reduction from baseline in the positive and negative symptom scale, indicating the possibility of acting on TAAR1 as a new class of psychotropic agent.

In this plethora of TAAR1 pro-cognitive indications, we investigated its capabilities on the early signs of neurodegeneration in models of Aβ toxicity. Soluble oligomers of Aβ-peptides represent a hallmark in AD pathogenesis [29]. One of the most well-known effects of Aβ oligomers is consistent damage to neurons and the consequent impact on synaptic transmission. The latter has a behavioural impact and induces slightly lower performance in learning and memory tasks [30]. There is abundant human neuropathological and experimental evidence that there is early functional and structural synaptic loss in AD, which relates to disease severity. Several studies in vitro and in experimental models indicate that Aβ oligomers are primarily responsible for glutamatergic synaptic dysfunction and dendritic pathology in AD, involving the interaction of NMDARs and mGluR5 receptors, triggering downstream cascades that disrupt LTP, promoting LTD and leading to reduced numbers and the collapse of dendritic spines [29]. In the early stages, Aβ oligomers elicit reversible synaptic changes. Later in the disease process, associated tau-related neuropathology triggers irreversible changes. There are many indications that Aβ may directly affect NMDA receptor function. For example, natural Aβ dodecameric oligomers co-immunoprecipitate with GluN1 and GluN2A [31]. Moreover, Aβ1–42 oligomers bind selectively to GluN1 and GluN2B expressing neurons but not GABAergic neurons [32]. Recent evidence indicates that the binding of Aβ to postsynaptic anchoring proteins such as PSD-95 is the cause of the Aβ1–42 effects on NMDA receptors [32,33]. Neuronal cortical cultures treated with soluble oligomers of Aβ1–40 (0.1–10 mM) have reduced levels of the synaptic PSD-95 and AMPA receptors in a concentration- and time-dependent manner [32]. Since this effect was prevented by the NMDA receptor channel blocker (+)MK-801, there is a strong indication of the possible direct activation of NMDA receptors by Aβ [34]. Additionally, synaptic NMDARs are localised to postsynaptic densities (PSDs), where they form a large macromolecular signalling complex of synaptic scaffolding and adaptor proteins which physically link the receptors to kinases, phosphatases and other downstream signalling proteins and to group I metabotropic glutamate receptors (mGluRs, [35]).

Specifically, NMDA dependent excitoxocity and GSK3 mediated Tau hyperphosporylation are crucial hallmarks of AD [22]. We could hypothesise that TAAR1 signalling inhibition has the potential to reduce hyperactivated NMDAR, thus reducing glutamate excitotoxicity which is highly implicated in the AD brain [22].

In general, changes in the functionality of NMDARs might be induced by, among others, modifications in cellular trafficking. This trafficking includes the movement of NMDARs between synaptic and extra-synaptic localisations at the surface of neurons and the cycle between intracellular compartments and the plasma membrane [36]. In 2005, Snyder and colleagues demonstrated that β-amyloid can regulate the surface expression and endocytosis of NMDA-type glutamate receptors, producing a rapid and persistent depression of NMDA-evoked currents in cortical neurons [25]. We observed a similar effect in our cortical cultures (Figure 1 and Figure 2) in control and TAAR1 KO mice. In our hands, TAAR1 treatment can increase the total cell surface, synaptic and non-synaptic expression of GluN1 and GluN2A subunits of the NMDA receptor. The interplay between TAAR1 and the glutamate system was already well known. The lack of TAAR1 causes the reduction of cortical glutamate NMDA receptor activity, indicating that TAAR1 can modulate PFC-related processes and behaviours [12]. The expression of GluN1 and GluN2B subunits of NMDA receptors was decreased in PFC with a concomitant decrease in the phosphorylation site on the GluN1 subunit (S896), while no significant alterations were found in AMPA receptor levels and phosphorylation. Here, we demonstrated a direct TAAR1 agonist effect on the glutamate system by showing that the TAAR1 agonist RO5256390 induces a rearrangement of NMDA receptors in vitro, thereby providing a plausible explanation for already mentioned effects on high-order PFC cognitive processes. It was reported that Aβ 1-42 could reduce the levels of the synaptic PSD-95 [25,34,35,36,37]. TAAR1 agonist seems to overcome such PSD-95 reduction, as shown in Figure 3. It has been shown that Aβ downregulates the levels of synaptophysin and PSD-95 in hippocampal cultures [37,38,39]. On the other hand, other authors reported increases or no changes in presynaptic protein expression levels in early and late-stage AD [38,39]. More recently, Doré et al. notably reported that PSD-95 protects synapses from Aβ toxicity, suggesting that low levels of synaptic PSD-95 may be a molecular sign indicating synapse vulnerability to Aβ [40]. Thus, the expression of presynaptic markers appears to differ with disease stage and may indicate greater vulnerability of postsynaptic versus presynaptic elements.

Several possibilities should be considered with respect to the mechanism by which TAAR1 could modulate cortical glutamate NMDA receptor-related transmission and functions. It is known that TAAR1 exerts a potent modulatory influence over the different monoaminergic systems and similar mechanisms could be involved in the regulation of cortical transmission. Further studies are necessary to explore all of these potential mechanisms.

The infusion of oligomeric Aβ (1–42) into the brain provides an excellent in vivo model, which recapitulates the amyloidopathy, and the consequent neuronal cell death. Firstly, Aβ deposits occur in the entorhinal and frontal cortices of AD patients and the amount of Aβ in these regions correlates with memory loss [41]. This method allows one to replicate the increase of Aβ peptides spatially and temporally, preventing any compensatory or side effects that may be encountered with transgenic lines [42].

In line with previous works, our results illustrated a significant impairment in spatial working memory in Aβ-treated mice. It was previously shown that brain Aβ1–42 administration in rodents causes apparent memory deficits that mimic the cognitive decline in AD [43,44]. Previous studies have already demonstrated that Aβ1–42 aggregates injected i.c.v. can be found in the hippocampus for weeks. Likewise, cognitive decline in AD patients is linked to elevated brain levels of Aβ, particularly neurotoxic Aβ1–42. We demonstrated that AD-related learning and memory deficits in an Aβ mouse model of AD could be ameliorated by a TAAR1 agonist (Figure 6). In addition, the locomotor activity and anxiety measure in the open field test did not differ between the groups. Thus, the observed impairment of memory of the mice cannot be attributed to the differences in their locomotion activities or anxiety levels.

The precise neural mechanism of the pro-cognitive effects of TAAR1 is not fully understood. TAAR1 is expressed in the PFC, a region that has a crucial role in executive functions and working memory [12,45]. Moreover, it has been shown that TAAR1 overexpression in cells significantly increased extracellular glutamate levels in the synaptic cleft, possibly through the decreased expression of excitatory amino acid transporter 2 (EAAT2) in astrocytes [46], suggesting the possibility of direct modulation of the glutamatergic system. However, further studies that explore the possible mechanisms underlying the precognitive effects of TAAR1 activation are needed.

Intriguingly, another member of the TAAR family, TAAR5, previously considered to be exclusively olfactory, was recently found in the limbic brain areas and its role in adult neurogenesis was demonstrated [11,47,48]. It would be of interest to explore whether TAAR5 or other TAARs could be involved in the regulation of cognitive function similarly to TAAR1.

## 4. Materials and Methods

### 4.1. Animals

All procedures involving animals and their care were carried out in accordance with the guidelines established by the European Community Council (Directive 2010/63/EU of 22 September 2010) and were approved by the Belgian Ministry of Health (Neurosciences LA1500024). Here, 3–5-month-old littermate C57Bl6 wild type (WT) and TAAR1 knockout (TAAR1-KO; generous gift of H. Lundbeck A/S) mice of both sexes were used in the experiments. The mice were housed 3–5/cage and maintained under standard lab conditions (12 h light/dark cycle, 21 ± 1 °C and 40–70% humidity) with food and water provided ad libitum. All experiments were conducted during the light phase. One hour before behavioural experiments, the mice were habituated to an experimental room.

### 4.2. Cell Cultures

Cortical neuronal cultures were obtained from E17-E18 embryos from TAAR1-KO and control animals, plated in 6-wells with a density of 1 × 10^6^ neurons/well (14DIV) and processed for Western blot (WB) with specific antibodies. Cell cultures were performed as follows: we collected embryos from anaesthetised pregnant mice and then removed brains from the skulls and put them in cold HBSS (Gibco, Aalst, Belgium, cat#11530476). We divided the hemispheres, removed the meninges and dissected the cortex and the hippocampi. Four hemi-cortexes were placed in 5 mL of trypsin 0.125% (Thermo Fisher Scientific, Aalst, Belgium, cat#25-200-056) + DNAse (Sigma Aldrich, Overijse, Belgium, cat#D5025-15KU) and incubated in a water bath at 37 °C for 30 min. After incubation, we added a few ml of complete Neurobasal Medium (Thermo Fisher Scientific, Aalst, Belgium cat#21103049) + 10% FBS to trypsin solution and centrifuged for 5 min at 1200 RPM. Supernatant was discarded and we added fresh Neurobasal + 10% foetal bovine serum (Thermo Fisher, Aalst, Belgium, cat#11550356). We then dissociated the tissue gently by pipetting. The solution was filtered with a cell strainer (40 µm pore size, Corning, Berlin, Germany, cat#352340) and centrifuged for 7 min at 700 RPM. Finally, the supernatant was discarded and the cells were resuspended in a complete neurobasal medium. Cells were plated at 1 × 10^6^ neurons/well; after 3 h, the medium was changed with fresh complete neurobasal medium. Drugs were added after 14 days in vitro as follows: Aβ 1-42, 1 µM (AnaSpec, Waddinxveen, The Netherlands, cat#AS-20276) vs. vehicle NH_4_OH_4_ (AnaSpec, Waddinxveen, The Netherlands, cat#AS-61322), 1 µM; TAAR1 full agonist RO5256390 (Sigma Aldrich, Overijse, Belgium, cat#ML1893-5MG) 1 µM vs. vehicle DMSO (Sigma Aldrich, Overijse, Belgium cat#D2650-100ML) or Aβ 1-42, 1 µM and (to evaluate the RO5256390 effect after) 1 h, TAAR1 full agonist RO5256390 1 µM. All of the other treatments were performed for 1 h at 37 °C.

### 4.3. Surface Biotinylation Assay

After drug application, neurons were placed on ice and rinsed in cold HBSS. Neurons were then incubated in HBSS containing 1.5 mg/mL sulpho-NHS-LC-biotin (Thermo Fisher, Aalst, Belgium, cat#21335) for 20 min at 4 °C. Neurons were rinsed twice for 5 min in HBSS + Lysine 200 mM (Alfa Aesar by Thermo Fisher Scientific, Aalst, Belgium, cat#A16249) and then lysed in 300 mL RIPA buffer (Millipore, Sigma Aldrich/Merck KGaA, Darmstadt, Germany, cat#20-188) with a complete protease inhibitor cocktail (Roche, Basel, Switzerland, cat#11836170001) + phosphatase inhibitor cocktail I (Abcam, Cambridge, UK, cat#ab201112). To determine the total protein concentration by immunoblotting, 10% of the cell lysate was removed (Input). To isolate biotinylated proteins, 80% of the cell lysate was incubated with NeutrAvidin agarose (50 mL; Thermo Fisher Scientific, Aalst, Belgium, cat#29201) overnight at 4 °C on the rotating wheel. Western blots were carried out and data were quantified by comparing the ratio of biotinylated to total protein for a given culture and normalising to control untreated cultures unless stated otherwise.

### 4.4. Western Blotting

Protein extracts were separated by 4–20% Mini-PROTEAN TGX Gels (Bio-Rad Laboratories, Temse, Belgium, cat#561091) and transferred to nitrocellulose membranes (GE Healthcare, Milan, Italy). Blots were immunostained overnight at 4 °C with the following primary antibodies: Purified anti-β-Amyloid (Biolegend, San Diego, CA, USA, cat#SIG-3920), anti-GluN2A (Merck Millipore, Hoeilaart, Belgium, cat#07-632), anti-GluN1 (Merck Millipore, Hoeilaart, Belgium, cat#05-432) and anti-PSD95 (Merck Millipore, Hoeilaart, Belgium, cat#MAB1598). After washing, the membranes were incubated for 2 h at room temperature with the appropriate secondary antibody (anti-mouse, GE Healthcare, Milan, Italy, cat#NXA931; anti-rabbit, cat#NA934V). Following secondary antibody incubations, membranes were washed and finally incubated with ECL detection reagent (Amersham, UK, cat#RPN2232) for 5 min.

Immunocomplexes were visualised by chemiluminescence using the LI-COR Biosciences Imaging System (LI-COR Biosciences, Bad Homburg vor der Höhe, Germany) and analysed using the Image Studio Lite software 5.2 (LI-COR Biosciences, Bad Homburg vor der Höhe, Germany).

### 4.5. AD In Vivo Model and Drugs Administration

Before the experimental procedures, mice were randomly divided into experimental groups (n = 12 for experimental group). The animal’s body weight variation was considered and counterbalanced across the groups. To develop the AD model in mice, we used a well-established method consisting of a direct Aβ injection as described previously [30]. Before the injection, Aβ1–42 protein was dissolved in NH_4_OH_4_ (1 µg/µL) in tubes incubated for 2 days at 37 °C and then vortexed several times to cause peptide aggregation. Next, anaesthetised mice (Ketamine 75 mg/kg + Medetomidine 1 mg/kg) were implanted in the right lateral ventricle with a stainless steel cannula by stereotaxic surgery (L ± 1.0; DV-3.0; AP (mm) − 0.22 from dura; cat#C315GMN, P1 technologies, Düsseldorf, Germany) through a hole drilled in the skull. One anchor screw (Centrostyle, Vedano Olona, Italy, cat#00374) and cement (Harvard Cement; Hoppegarten, Germany, cat#700221) were used to attach the guide cannula to the skull. Sutures were used to close the incision, a topical antibiotic was applied and a dummy was inserted into the guide cannula to prevent clogging (P1 technologies, Düsseldorf, Germany, cat#C315DCMN/Spc). Mice recovered from the surgery for at least 7 days before injection and 7 more days before behavioural testing started.

Mice received aggregated Aβ1–42 protein (3 µg/µL) or NH_4_OH_4_ (Aβ1–42 vehicle, 3 µL) injections through a guide cannula system (P1 technologies, Düsseldorf, Germany, cat# C315GMN/Spc) at a rate of 1 µL/min. The cannula was removed after 3 min to minimise backflow. After the injection, mice were placed on a thermal pad until they awakened.

To investigate whether a TAAR1 agonist treatment would provide protection against Aβ1–42 peptide-induced AD, 2 weeks after the Aβ injection, mice were treated with TAAR1 agonist or vehicle (respectively 3 µg/µL or 3 µL). Mice recovered 7 days before behavioural testing started.

### 4.6. Y-Maze Task

Spontaneous alternation is a measure of spatial working memory. Such short-term working memory was assessed by recording spontaneous alternation behaviour during a single session in the Y-maze. Each mouse was placed at the end of one arm and allowed to move freely through the maze during a 5 min session. The series of arm entries were camera recorded (EthoVision XT, Noldus Instruments, Wageningen, The Netherlands). An arm choice was considered only when both forepaws and hind paws fully entered into the arm. The Y-maze was cleaned after each test with 70% alcohol to remove any residual odours. Alternation was defined as a successive entrance onto the three different arms. The number of correct entrance sequences (e.g., ABC, BCA) was defined as the number of actual alternations. The number of total possible alternations was therefore the total number of arm entries minus two and the percentage of alter-nation was calculated as actual alternations/total alternations × 100.

### 4.7. Locomotor Activity

Locomotor activity was evaluated as described before [49] under illuminated conditions. Animals were tested individually for defined periods with 10 min intervals. The total distance travelled was measured and expressed in terms of centimetres travelled by the animal using EthoVision generated data (Noldus Instruments, Wageningen, The Netherlands) as well as velocity, expressed in terms of cm/second.

### 4.8. Statistical Analysis

The results obtained were expressed as the mean ± SEM. Statistical analysis was performed by using one-way ANOVA followed by Dunnett’s post-test when comparing more than two groups. In some cases, one sample t-test was used to evaluate significance against the hypothetical zero value. Statistical analysis was performed by using Graph Pad Prism 4.0 software (San Diego, CA, USA). Data were considered statistically significant when a value of *p* < 0.05 was achieved.

## 5. Conclusions

In conclusion, the present study showed the role of TAAR1 in modulating NMDA receptor expression and localization and suggests that TAAR1 agonists might be promising tools to counteract cognitive dysfunction in disorders involving amyloid accumulation, such as AD.

## Figures and Tables

**Figure 1 ijms-23-07811-f001:**
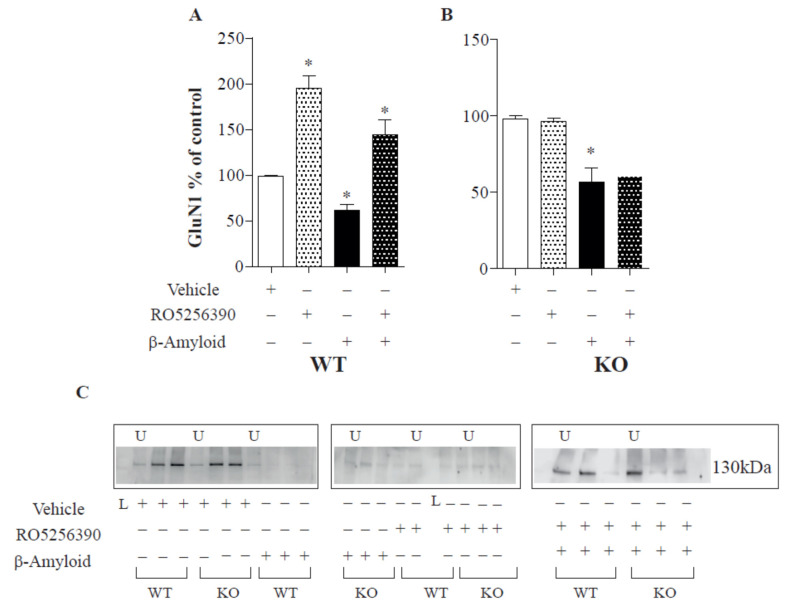
Effect of Aβ and RO5256390 on NR1 receptors in vitro. Aβ application in cell medium reduces NR1 surface expression, while TAAR1 activation (induced by RO5256390 agonism) is able to counteract Aβ effects in control cells (**A**) but not in KO cell cultures (**B**). (**C**) Examples of Western blot images: U = unbound (cells not treated with biotin), L = ladder. Results are representative of six different cell cultures (* *p* < 0.05).

**Figure 2 ijms-23-07811-f002:**
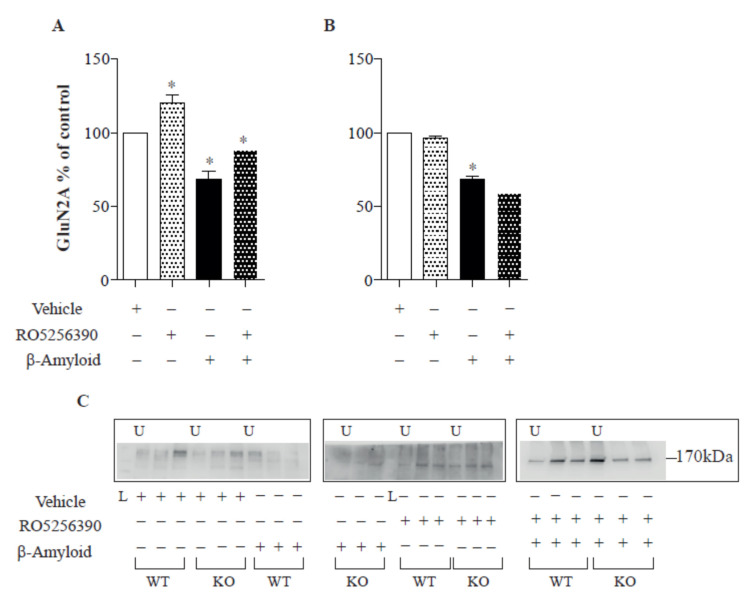
Effect of Aβ and RO5256390 on GluN2A receptors in vitro. Aβ application in cell medium reduces GluN2A surface expression, while TAAR1 activation (induced by RO5256390 agonism) is able to counteract Aβ effects in control cells (**A**) but not in KO cell cultures (**B**). (**C**) Examples of Western blot images: U = unbound, L = ladder. Results are representative of six different cell cultures (* *p* < 0.05).

**Figure 3 ijms-23-07811-f003:**
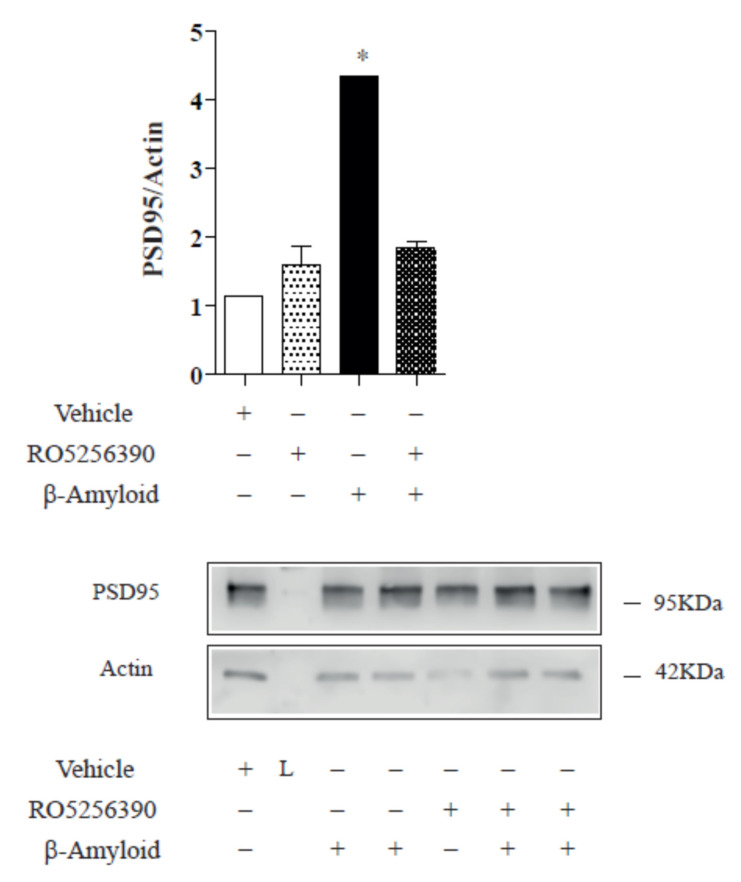
TAAR1 activation induces an increase in PSD-95 protein expression. The TAAR 1 agonist demonstrated the ability to increase the PSD amount in control cortical cells 14 DIV. Representative Western blot images. Results are representative of three different cell cultures (* *p* < 0.05).

**Figure 4 ijms-23-07811-f004:**
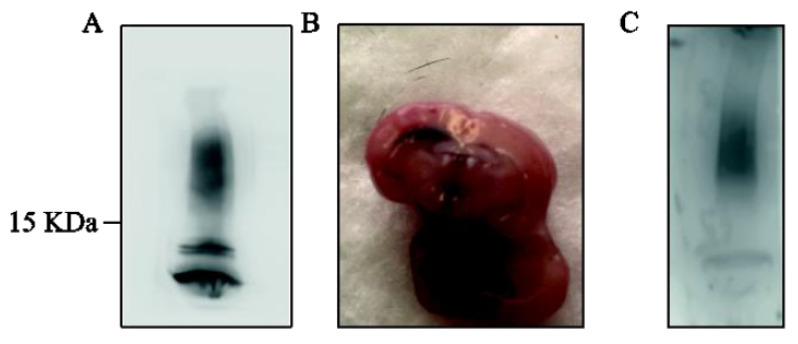
(**A**) Immunoblot with anti-b-amyloid antibody (Biolegend SIG-3920) showed small-sized Aβ monomer, dimer and tetramer bands. Aβ was aggregated for 24 h at 37 °C. (**B**) The effective distribution of the injected solution (methylene blue). (**C**) Immunoblot with anti-b-amyloid antibody (Biolegend SIG-3920) on hippocampal protein extracts after 1 week of Aβ i.c.v. injection showed the Aβ in the mouse brain. Results are representative of three different injections.

**Figure 5 ijms-23-07811-f005:**
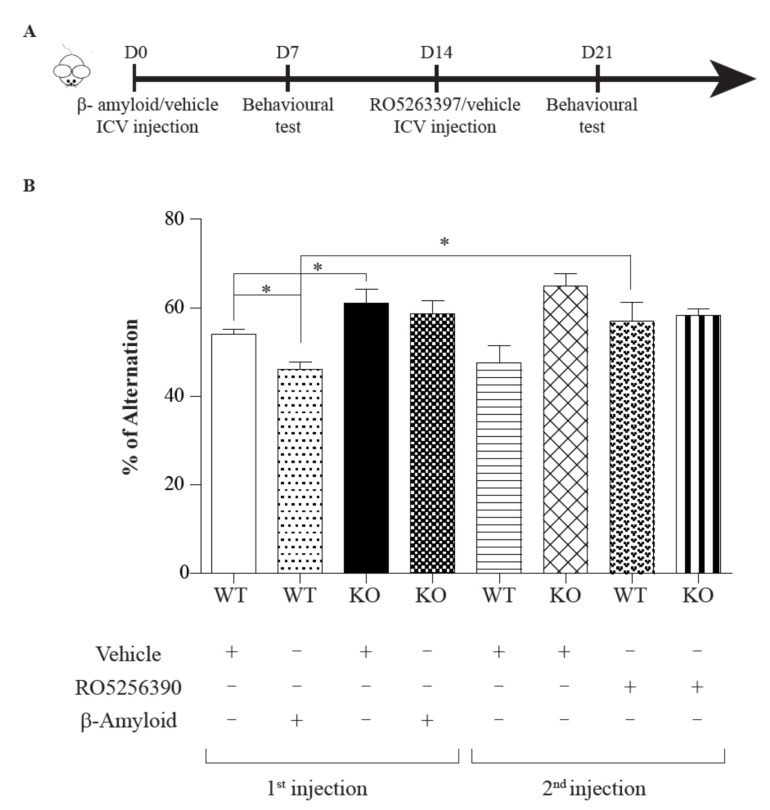
(**A**) Protocol of the Aβ-induced mouse model and behavioural analysis (n = 12 mice/group). (**B**) ICV Aβ infusion is able to induce a slight cognitive impairment in control animals while the KO seem to be more resistant to Aβ effects (* *p* < 0.05). Furthermore, neither the vehicle nor the TAAR1 agonist administration influenced the Y-maze performance in both control and knockout animals.

**Figure 6 ijms-23-07811-f006:**
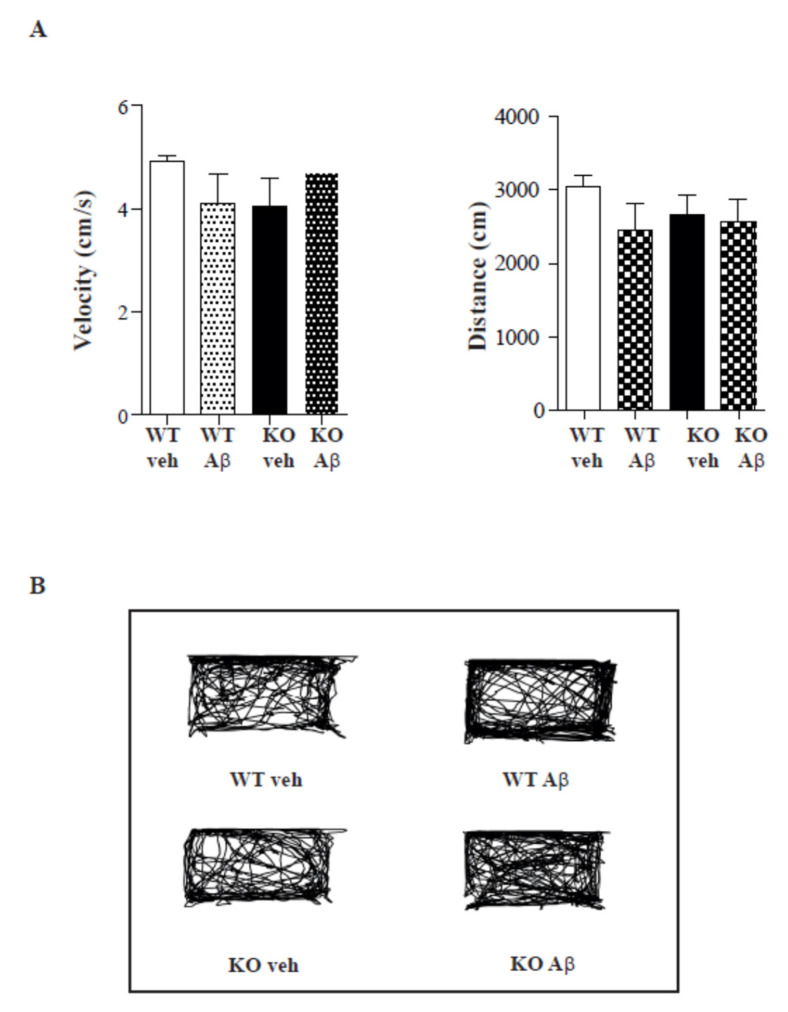
(**A**) Open field velocity and distance for TAAR1 and control animals treated or not with Aβ. (**B**) Trace examples of thigmotaxis analysis in a square open field. Neither control nor TAAR1 mice showed anxiety, regardless of the treatment received (Aβ or vehicle).

## Data Availability

Not applicable.

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
