# Peer review of "Trace Amine Associate Receptor 1 (TAAR1) as a New Target for the Treatment of Cognitive Dysfunction in Alzheimer’s Disease"

_ijms, 2022, doi:10.3390/ijms23147811_

Round 1

Reviewer 1 Report

1. Remove PMID: 28723415

2. Figure 1 C, WB is not clear, replace it with good one, also share fullblot with ladder markings

3. Figure 2C, WB is not clear, replace it with good one, also share fullblot with ladder markings

4. I think PSD 95 is not a doube band

5.  Figure legends should have more information with experiments number

6. Mention appropriate cat# for all cosumables

7. Mention the amount proetin loaded on WB

8. Avoid pronouns/adverbs in the begiing of the senetence

Author Response

Please see tha attachment

Reviewer 2 Report

I have some comments on the manuscript.

  1. Rephrase “It is believed that Beta-Amyloid peptide (Aβ) is causative in the disease process”. In the abstract section.
  2. Write the limitations of your study at the end of the abstract section.
  3. Elaborate on the last paragraph of the introduction section.
  4. Discuss and Cite relevant PMID: 34970114, PMID: 34211570, PMID: 32462551, PMID: 30605887 and PMID: 30503937 in your manuscript.
  5. Ave you found any significant difference between males and females concerning different parameters.
  6. Write the exact p-value in all the bar diagrams.
  7. The significance value is missing in figure no 6.
  8. Elaborate in the discussion section by citing recent and relevant articles as per your manuscript.
  9. Histology of the associated brain areas in AD and other molecular level studies will be needed to justify the role of TAAR1 as a new target for the treatment of cognitive dysfunction in Alzheimer's disease.
  10. Complete editorial checking will be needed to correct the grammatical and punctuation mistakes.

Round 2

Reviewer 2 Report

Revised manuscript is suitable for publication. 

Author Response

Dear Editor, Dear Reviewer,

We would like to thank you for the useful suggestions, we feel that the manuscript is improved.

In the reviewer's comment we still see the request for an English revision. We sent our manuscript to a proof reading service, as you can see in the attched document.

Best,

Damiana Leo

This manuscript is a resubmission of an earlier submission. The following is a list of the peer review reports and author responses from that submission.